# Diversity-Driven Exploration Strategy for Deep Reinforcement Learning

**Zhang-Wei Hong, Tzu-Yun Shann, Shih-Yang Su, Yi-Hsiang Chang, Tsu-Jui Fu, and Chun-Yi Lee**

Department of Computer Science, National Tsing Hua University
`{williamd4112,arielshann,at7788546,shawn420,rayfu1996ozig,cylee}`
`@gapp.nthu.edu.tw`

## Abstract

Efficient exploration remains a challenging research problem in reinforcement learning, especially when an environment contains large state spaces, deceptive or sparse rewards. To tackle this problem, we present a diversity-driven approach for exploration, which can be easily combined with both off- and on-policy reinforcement learning algorithms. We show that by simply adding a distance measure regularization to the loss function, the proposed methodology significantly enhances an agent's exploratory behavior, and thus prevents the policy from being trapped in local optima. We further propose an adaptive scaling strategy to enhance the performance. We demonstrate the effectiveness of our method in huge 2D gridworlds and a variety of benchmark environments, including Atari 2600 and MuJoCo. Experimental results validate that our method outperforms baseline approaches in most tasks in terms of mean scores and exploration efficiency.

## 1 Introduction

In recent years, deep reinforcement learning (DRL) has attracted attention in a variety of application domains, such as game playing [1, 2] and robot navigation [3]. However, exploration remains a major challenge for environments with large state spaces, deceptive local optima, or sparse reward signals. In an environment with deceptive rewards, an agent can be trapped in local optima, and never discover alternate strategies to find larger payoffs. For example, in *HalfCheetah* of MuJoCo [4], the agent quickly learns to flip on its back and then "wiggles" its way forward, which is a sub-optimal policy [5]. In addition, environments with only sparse rewards provide few training signals, making it hard for agents to discover feasible policies. A common approach for exploration is to adopt simple heuristic methods, such as $\epsilon$-greedy [1, 6] or entropy regularization [7]. However, such strategies are unlikely to yield satisfactory results in tasks with deceptive or sparse rewards [8, 9]. A more sophisticated line of methods provides agents with bonus rewards whenever they visit a novel state. For example, [10] uses information gain as a measurement of state novelty. In [11], a counting table is used to estimate the novelty of a visited state. Neural density model [12] has also been utilized to measure bonus rewards for agents. In [13, 14], the novelty of a state is estimated from the prediction errors of their system dynamics models. These methods, however, often require statistical or predictive models to evaluate the novelty of a state, and therefore increase the complexity of the training procedure.

In order to deal with the issue of complexity, a few researchers attempts to embrace the idea of random perturbation from evolutionary algorithms [5, 8]. By adding random noise to the parameter space, their methods allow RL agents to perform exploration more consistently without introducing extra computational costs. Despite their simplicity, these methods are less efficient in large state spaces, as random noise changes the behavioral patterns of agents in an unpredictable fashion [5, 8]. In [15], the authors propose to address the problem by training multiple value functions with bootstrap subsamples. However, it requires extra model parameters, causing additional computational overheads.

In this paper, we present a diversity-driven exploration strategy, a methodology that encourages a DRL agent to attempt policies different from its prior policies. We propose to use a distance measure to modify the loss function to tackle the problems of large state spaces, deceptiveness and sparsity in reward signals. The distance measure evaluates the novelty between the current policy and a set of prior policies. Our method draws inspiration from novelty search [16–18], which promotes population-based exploration by encouraging novel behaviors. Our method differs from it in several aspects. First, we cast the concept of novelty search from evolution strategies into DRL frameworks. Second, we train a single agent instead of a population. Third, novelty search ignores rewards altogether, while ours optimizes the policy using both the reward signals and the distance measure. A work parallel to ours similarly employs novelty search for exploration [19]. However, their method still lies in the realm of genetic algorithms. We demonstrate that our methodology is complementary and easily applicable to most off- and on-policy DRL algorithms. We further propose an adaptive scaling strategy, which dynamically scales the effect of the distance measure for enhancing the overall performance. The adaptive scaling strategy consists of two methods: a distance-based method and a performance-based method. The former method adjusts the scaling factor based on the distance measure, while the latter method scales it according to the performances of the prior policies.

To validate the effectiveness of the proposed diversity-driven exploration strategy, we first demonstrate that our method does lead to better exploratory behaviors in 2D gridworlds with deceptive or sparse rewards. We compare our method against a number of contemporary exploration approaches (i.e., the baselines). Our experimental results show that while most baseline agents employing the contemporary exploration approaches are easily trapped by deceptive rewards or fail in the sparse reward settings, the agents employing our exploration strategy are able to overcome aforementioned challenges, and learn effective policies even when the reward signals are sparse or when the environments contain deceptive rewards. We have further evaluated our method in a variety of benchmark environments, including Atari 2600 [20] and MuJoCo [4]. We performed various experiments to demonstrate the benefits of diversity-driven exploration strategy. We show that the proposed methodology is superior to, or comparable with the baselines in terms of mean scores and learning time. Moreover, we provide a comprehensive ablative analysis of the proposed adaptive scaling strategy, and investigate the impact of different scaling methods on the learning curves of the DRL agents.

The main contributions of this paper are summarized as follows:

- A simple, effective, and efficient exploration strategy applicable to most off- and on-policy DRL algorithms.
- A promising way to deal with large state spaces, deceptive rewards, and sparse reward settings.
- A loss function designed for encouraging exploration by the use of a distance measure between the current policy and a limited set of the most recent policies.
- An adaptive scaling strategy consisting of two scaling methods for the distance measure. It enhances the overall performance.
- A comprehensive comparison between the proposed methodology and a number of contemporary approaches, evaluated on three different environments.

The remainder of this paper is organized as the following. Section 2 provides background material. Section 3 walks through the proposed exploration strategy in detail. Section 4 presents the experimental setup and results. Section 5 discusses the related work. Section 6 concludes this paper.

## 2 Background

In this section, we review the concept of RL, and the off- and on-policy methods used in this paper.

### 2.1 Reinforcement Learning

RL is a method to train an agent to interact with an environment $\mathcal{E}$. An RL agent observes a state $s$ from the state space $\mathcal{S}$ of $\mathcal{E}$, takes an action $a$ from the action space $\mathcal{A}$ according to its policy $\pi(a|s)$, and receives a reward $r(s, a)$. $\mathcal{E}$ then transits to a new state $s'$. The agent's objective is to maximize its discounted accumulated rewards $G_t = \sum_{\tau=t}^{T} \gamma^{\tau-t} r(s_\tau, a_\tau)$, where $t$ is the current timestep, $\gamma \in (0, 1]$ the discount factor, and $T$ the horizon. The action-value function (i.e., Q-function) of a

given policy $\pi$ is defined as the expected return starting from a state-action pair $(s, a)$, expressed as $Q(s, a) = \mathbb{E}\big[G_t | s_t = s, a_t = a, \pi\big]$.

## 2.2 Off-Policy Methods

Off-policy methods decouple the behavior and target policies, enabling an agent to learn using samples collected by arbitrary policies or from an experience replay [1]. We briefly review two representative off-policy methods, namely Deep Q-Network (DQN) [1] and Deep Deterministic Policy Gradient (DDPG) [21].

**DQN.** DQN is a deep neural network (DNN) parameterized by $\theta$ for approximating the optimal Q-function. For exploration, it follows an $\epsilon$-greedy strategy described in [1]. The network is trained with samples drawn from an experience replay $Z$, and is updated according to the loss function $L_{DQN}$ expressed as:

$$L_{DQN} = \mathbb{E}_{s,a,r,s' \sim U(Z)}\big[(y - Q(s, a, \theta))^2\big], \tag{1}$$

where $y = r(s, a) + \gamma \max_{a'} Q(s', a', \theta^-)$, $U(Z)$ is a uniform distribution over $Z$, and $\theta^-$ the parameters of the target network. $\theta^-$ is updated by $\theta$ at predefined intervals.

**DDPG.** DDPG is an actor-critic approach based on the deterministic policy gradient (DPG) algorithm [22] that learns policies over continuous action spaces. The critic estimates the Q-function similarly as that of DQN, with a minor modification to $y$, expressed as $y = r(s, a) + \gamma Q(s', \pi(s'))$. The actor is trained to maximize the critic's estimated Q-values, with a loss function $L_{actor}$ given by:

$$L_{actor} = -\mathbb{E}_{s \sim \mathcal{Z}}\big[Q(s, \pi(s))\big], \tag{2}$$

where $s$ is sampled from $Z$. DDPG uses a stochastic policy $\hat{\pi}(s) = \pi(s) + N$ for exploration, where $N$ is a noise process. $N$ can be either normally distributed or generated by the Ornstein-Uhlenbeck (OU) process [21].

## 2.3 On-Policy Methods

On-policy methods update their value functions based on the samples generated by the current policy. We review a state-of-the-art on-policy method called Advantage Actor-Critic (A2C) [7, 23], which is evaluated and compared to the proposed methodology in Section 4. A2C is a synchronous variant of Asynchronous Advantage Actor-Critic (A3C) [7], which trains agents in parallel, on multiple instances of the environment. A2C offers better utilization of GPUs than A3C. Similarly, the critic estimates the value function $V(s)$. The actor optimizes $\pi(a|s, \theta)$ by minimizing the following loss function:

$$L_{actor} = -\mathbb{E}_{s,a \sim \pi}\big[G_t - V(s) + \beta H(\pi(.|s, \theta))\big], \tag{3}$$

where $\beta$ is a hyperparameter for controlling the strength of the entropy $H(\pi(.|s, \theta))$. A2C utilizes $H(\pi(.|s, \theta))$ to encourage exploratory behaviors, as well as prevent agents from converging prematurely to sub-optimal policies.

# 3 Diversity-Driven Exploration Strategy

The main objective of the proposed diversity-driven exploration strategy is to encourage a DRL agent to explore different behaviors during the training phase. Diversity-driven exploration is an effective way to motivate an agent to examine a richer set of states, as well as provide it with an approach to escape from sub-optimal policies. It can be achieved by modifying the loss function $L_D$ as follows:

$$L_D = L - \mathbb{E}_{\pi' \in \Pi'}[\alpha D(\pi, \pi')], \tag{4}$$

where $L$ indicates the loss function of any arbitrary DRL algorithms, $\pi$ is the current policy, $\pi'$ is a policy sampled from a limited set of the most recent policies $\Pi'$, $D$ is a distance measure between $\pi$ and $\pi'$, and $\alpha$ is a scaling factor for $D$. The second term in Eq. (4) encourages an agent to update $\pi$ with gradients towards directions such that $\pi$ diverges from the samples in $\Pi'$. Eq. (4) provides several favorable properties. First, it drives an agent to proactively attempt new policies, increasing the opportunities to visit novel states even in the absence of reward signals from $\mathcal{E}$. This property is especially useful in sparse reward settings, where the reward is zero for most of the states in

$S$. Second, the distance measure $D$ motivates exploration by modifying an agent's current policy $\pi$, instead of altering its behavior randomly. Third, it allows an agent to perform either greedy or stochastic policies while exploring effectively in the training phase. For greedy policies, since $D$ requires an agent to adjust $\pi$ after each update, the greedy action for a state may change accordingly, potentially directing the agent to explore unseen states. This property also ensures that the agent acts consistently in the states it has been familiar with, as $\pi$ and $\pi'$ yield the same outcomes for those states. These three properties allow a DRL agent to explore an environment in a systematic and consistent manner. The choice of $D$ can be KL-divergence, L2-norm, or mean square error (MSE). The interested reader is referred to our supplementary material for more details of how the distance measure is selected. In the subsequent sections, we explain how diversity-driven exploration can be combined with off-policy and on-policy methods, for both discrete and continuous control problems.

### 3.1 Implementation on Off-Policy Methods

Most off-policy DRL algorithms [1, 21] adopt the experience replay mechanism to stabilize the learning process. We show that diversity-driven exploration can be readily applied to existing algorithms for both discrete (DQN) and continuous (DDPG) control tasks, with only a few modifications to the experience replay buffer $Z$.

**Div-DQN.** We make the following changes to the DQN algorithm. First, we additionally store the past Q-values (denoted as $Q'(s,a)$) in $Z$. Second, for the sake of defining a proper distance measure, we use a probabilistic formulation as in [5] by applying the softmax function over the predicted Q-values. We therefore define $\pi(a|s) = \exp(Q(s,a))/\Sigma_{a'\in\mathcal{A}}\exp(Q(s,a'))$. $\pi'(a|s)$ is defined similarly but uses $Q'(s,a)$ instead. We adopt KL-divergence as the distance measure, denoted as $D_{KL}$. Eq. (4) is rewritten as:

$$L_D = L - \mathbb{E}_{\hat{Q}(s,a)\sim U(Z)}[\alpha D_{KL}(\pi'(.|s)||\pi(.|s))], \tag{5}$$

where $\alpha$ can be either a predefined value, or determined by the adaptive methods in Section 3.3.

**Div-DDPG.** For diversity-driven DDPG, we use the actions stored in $Z$, and modify Eq. (4) as:

$$L_D = L - \mathbb{E}_{s,a'\sim U(Z)}[\alpha D(\pi(s), a')], \tag{6}$$

where $a'$ is a prior action sampled from $Z$. The distance measure $D$ here is simply an MSE function between $\pi(s)$ and $a'$. The value of $\alpha$ is determined in a similar fashion as that of DQN. Please note that we do not use the stochastic exploration policy mentioned in Section 2.2, since $D$ alone is sufficient enough for improving the exploratory behavior.

### 3.2 Implementation on On-Policy Methods

Apart from off-policy methods, we explain how our methodology can be applied to on-policy methods.

**Div-A2C.** As A2C has no experience replay, we maintain the $n$ most recent policies for calculating the distance measure $D$. In general cases, $n = 5$ is sufficient to yield satisfactory performance. The loss function $L_D$ is thus expressed as:

$$L_D = L - \mathbb{E}_{s\sim\tau}[\mathbb{E}_{\pi'\sim\prod}[\alpha_{\pi'}D_{KL}(\pi'(.|s)||\pi(.|s))]], \tag{7}$$

where $\tau$ represents the batch of on-policy data, $\prod$ is the set of prior polices, and $\pi'$ is a prior policy. KL-divergence is used as the distance measure. We describe in detail a performance-based method to scale $\alpha_{\pi'}$ for each individual prior policy $\pi'$ in Section 3.3.

### 3.3 Adaptive Scaling Strategy

Although $\alpha$ can be linearly annealed over time, we find this solution less than ideal in some cases. We demonstrate these cases in experimental results provided in Section 4. To update $\alpha$ in a way that leads to better overall performance, we consider two adaptive scaling methods: the distance-based and the performance-based methods. In our experiments, the off-policy algorithms use only the distance-based method, while the on-policy algorithm uses both methods for scaling $\alpha$.

**Distance-based.** Similar to [5], we relate $\alpha$ to the distance measure $D$. We adaptively increase or decrease the value of $\alpha$ depending on whether $D$ is below or above a certain threshold $\delta$. The simple approach we use to update $\alpha$ for each training iteration is defined as:

$$\alpha := \begin{cases} 1.01\alpha, & \text{if } \mathbb{E}\big[D(\pi, \pi')\big] \leq \delta \\ 0.99\alpha, & \text{otherwise} \end{cases}. \tag{8}$$

Please note that different values of $\delta$ are applied to different methods in our experiments. The method for determining the value of $\delta$ is described in the supplementary material.

**Performance-based.** While the distance-based scaling method is straightforward and effective, it alone does not lead to the same performance for on-policy algorithms. The rationale behind this is that we only use the five most recent policies ($n = 5$) to compute $L_D$, which often results in high variance, and instability during the training phase. Off-policy algorithms do not suffer from this issue, as they can utilize experience replay to provide a sufficiently large set of past policies. Therefore, we propose to further adjust the value of $\alpha$ for on-policy algorithms according to the performance of past policies to stabilize the learning process. We define $\alpha_i$ in either one of the following two strategies:

$$\alpha_{\pi'} := -(2(\frac{P(\pi') - P_{min}}{P_{max} - P_{min}}) - 1) \text{ (Proactive)}; \quad \alpha_{\pi'} := 1.0 - \frac{P(\pi') - P_{min}}{P_{max} - P_{min}} \text{ (Reactive)}, \tag{9}$$

where $P(\pi')$ denotes the average performance of $\pi'$ over five episodes, and $P_{min}$ and $P_{max}$ represent the minimum and maximum performance attained by the set of past policies $\Pi'$. Note that $\alpha_{\pi'}$ falls in the interval $[-1, 1]$ for the proactive strategy, and $[0, 1]$ for the reactive one. The proactive strategy incentivizes the current policy $\pi$ to converge to the high-performing policies in $\Pi'$, while keeping away from the poor ones. On the other hand, the reactive strategy only motivates $\pi$ to stay away from the underperforming policies. We provide a comprehensive ablative analysis for these strategies in Section 4. Note that we apply both Eq. (8) and Eq. (9) to the on-policy methods in our experiments.

### 3.4 Clipping of Distance Measure

In some cases, the values of $D$ become extraordinarily high, causing instability in the training phase and thus degrading the performance. To stabilize the learning process, we clip $D$ to be between $-c$ and $c$, where $c$ is a predefined constant. Please refer to our supplementary material for more details.

## 4 Experiments

In this section, we present our experimental results and discuss their implications. We first provide an overview of the experimental setup and the environments we used to evaluate our models in Section 4.1. In Sections 4.2~4.4, we report results in three different environments, respectively. We further provide an ablative analysis for the proposed methodology in the supplementary material.

### 4.1 Experimental Setup

#### 4.1.1 Environments

**Gridworld.** To provide an illustrative example of our method's effectiveness, we create a huge 2D gridworld (Fig. 1) with two different settings: (1) sparse reward setting, and (2) deceptive reward setting. In both settings, the agent starts from the top-left corner of the map, with an objective to reach the bottom-right corner to obtain a reward of 1. At each timestep, the agent observes its absolute coordinate, and chooses from four possible actions: *move north*, *move south*, *move west*, and *move east*. An episode terminates immediately after a reward is collected. In the deceptive reward setting illustrated in Fig. 1 (a), the central area of the map is scattered with small rewards of 0.001 to distract the agent from finding the highest reward in the bottom-right corner. On the other hand, in the sparse reward setting depicted in Fig. 1 (b), there is only a single reward located at the bottom-right corner.

**Atari 2600.** For discrete control tasks, we perform experiments in the Arcade Learning Environment (ALE) [20]. We select eight games varying in their difficulty of exploration according to [24]. In each game, the agent receives $84 \times 84 \times 4$ stacked grayscale images as inputs, as described in [1].

Table 1: Evaluation results of the gridworld experiments.

| | Deceptive Reward | | | | Sparse Reward | | |
|---|---|---|---|---|---|---|---|
| | $50 \times 50$ | $100 \times 100$ | $200 \times 200$ | | $50 \times 50$ | $100 \times 100$ | $200 \times 200$ |
| Vanilla-DQN | 0.010 | 0.010 | 0.010 | Vanilla-DQN | 0.300 | 0.100 | 0.000 |
| Noisy-DQN | 0.009 | 0.004 | 0.010 | Noisy-DQN | 0.400 | 0.200 | 0.000 |
| Div-DQN | 0.202 | 0.604 | 0.208 | Div-DQN | 1.000 | 1.000 | 1.000 |

**MuJoCo.** For continuous control tasks, we conduct experiments in environments built on the MuJoCo physics engine [4]. We select a number of robotic control tasks to evaluate the performance of the proposed methodology and the baseline methods. In each task, the agent takes as input a vector of physical states, and generates a vector of action values to manipulate the robots in the environment.

### 4.1.2 Baseline Methods

The baseline methods (or simply "baselines") adopted for comparison vary within different environments. For discrete control tasks, we select vanilla DQN [1], vanilla A2C [7], as well as their noisy net [8] and Curiosity-driven [14] variants (denoted by Noisy-DQN/A2C and Curiosity-DQN/A2C, respectively) as the baselines. For continuous control tasks, vanilla DDPG [21] and its parameter noise [5] variant (referred to as parameter noise DDPG) are taken as the baselines for comparison. All of these baselines are implemented based on *OpenAI Baselines*[1]. For each method, we adopt the setting that yields the highest overall performance during hyperparameter search. Our hyperparameter settings are provided in the supplementary material. Please note that for DQN and A2C, we choose their noisy net variants for comparison instead of the parameter noise ones, as the former variants lead to relatively better performance. On the other hand, we select parameter noise DDPG as our baseline rather than the noisy net variant, as the authors of [8] do not provide their implementations.

### 4.2 Exploration in Huge Gridworld

In this experiment, we evaluate different methods in 2D gridworlds with three different sizes: $50 \times 50$, $100 \times 100$, and $200 \times 200$. We consider only vanilla DQN and Noisy-DQN in this experiment. We report the performance of each method in terms of their average rewards in Table. 1, where the average rewards are evaluated over the last ten episodes. We plot the state-visitation counts of all methods (Fig. 2) on $200 \times 200$ gridworlds, in order to illustrate how agents explore the state space.

**Deceptive reward.** Fig. 1 (a) illustrates the deceptive gridworld. As shown in Table. 1, Div-DQN outperforms both vanilla and Noisy-DQN in this setting. From the state-visitation counts (Fig. 2 (a)(b)(c)), it can be observed that baseline methods are easily trapped in the area near the deceptive rewards, and have never visited the optimal reward in the bottom-right corner. On the other hand, it can be seen from Fig. 2 (c) that Div-DQN is able to escape from the area of deceptive rewards, explores all of the four sides of the gridworld, and successfully discovers the optimal reward of one.

**Sparse reward.** Fig. 1 (b) illustrates the sparse gridworld. From Table. 1, we notice that the average rewards of Noisy-DQN increase slightly as compared to those in the deceptive reward setting. As the size of gridworld increases, it fails to find the location of the reward. In contrast, Div-DQN consistently achieves 1.0 mean reward for all the gridworld sizes. In addition, from Fig. 2 (d), it can be seen that DQN spends most of its time wandering around the same route. Thus, its search range covers only a small proportion of the state space. Noisy-DQN explores a much broader area of the state space. However, the bright colors in Fig. 2 (e) indicates that Noisy-DQN wastes significant amount of time visiting explored states. On the other hand, Div-DQN is the only method that is capable of exploring the gridworld uniformly and systematically, as illustrated in Fig. 2 (f). These results validate that our methodology is superior in exploring large state spaces with sparse rewards.

Based on the above results, we conclude that the proposed diversity-driven exploration strategy does offer advantages that are favorable in exploring large gridworlds with deceptive or sparse rewards.

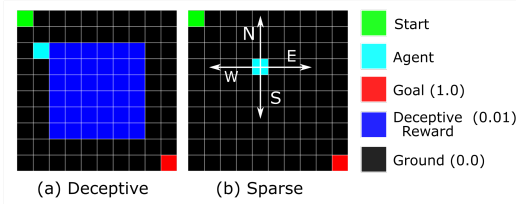

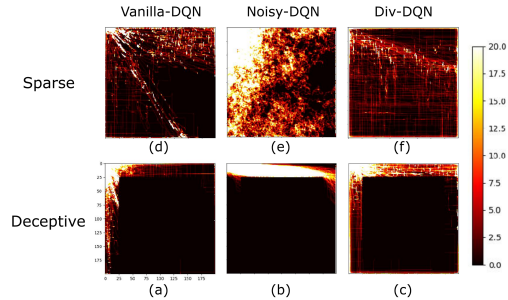

Figure 1: Gridworlds.      Figure 2: State-visitation counts of the gridworlds.

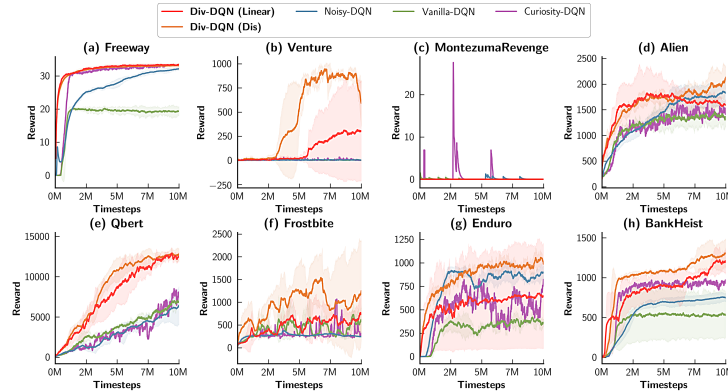

Figure 3: Comparison of learning curves for different DQN variants in Atari 2600.

## 4.3 Performance Comparison in Atari 2600

In addition to the gridworld environments, we evaluate our methodology in a more challenging set of environments *Atari 2600*. We provide an in-depth analysis for the empirical results of a few games selected from both the hard and easy exploration categories, according to the taxonomy in [24]. Each agent is trained with 40M frames, and the performance is evaluated over three random seeds. In Figs. 3 and 4, we plot the in-training median scores, along with the interquartile range. Div-DQN (Linear) and Div-DQN (Dis) correspond to our diversity-driven DQN with the linear decay and the distance-based methods for scaling $\alpha$, respectively. Div-A2C (Pro) and Div-A2C (Rea) correspond to our diversity-driven A2C with the proactive and the reactive methods in Sec. 3.3, respectively.

### 4.3.1 Hard Exploration Games

Fig. 3 plots the learning curves of all of the models in the training phase. It can be seen that our methods demonstrate superior or comparable performance to the baseline methods in all games. Particularly, we observe that our strategy helps an agent explore a wider area more efficiently compared to the other baselines, which is especially useful when the state spaces become sufficiently large. For example, in *Freeway*, we notice that our agents are able to quickly discover the only reward at the other side of the road, while the other methods remain at the starting position. This observation is consistent with the learning curves illustrated in Figs. 3 (a) and 4 (a), where Div-DQN and Div-A2C learns considerably faster and better than the baseline methods. In addition to the direct benefits of efficient and thorough exploration mentioned above, we also observe that the exploratory behaviors induced by our methods are more systematic, motivating our agents explore unvisited states. As shown in Fig. 3 (b), Div-DQN is the only one that learns a successful policy in *Venture*. On the contrary, as the vanilla DQN, A2C, Noisy-DQN/A2C, and Curiosity-DQN/A2C agents explore in a random way, they often bump into monsters they have previously seen and are quickly killed. It should be noted that although Div-A2C does not demonstrate a substantial increase in performance, it can be seen in Table S1 that it does end up with higher rewards than the baselines. We also observe that our method helps an agent ignore known small (deceptive) rewards, and discover alternative ways to obtain the optimal reward. For instance, in *Alien*, our agents learn to collect rewards while avoiding aliens by detouring, while the baselines focus on the immediate rewards in front of them without taking aliens into consideration. This enables our agents obtain higher average rewards than the

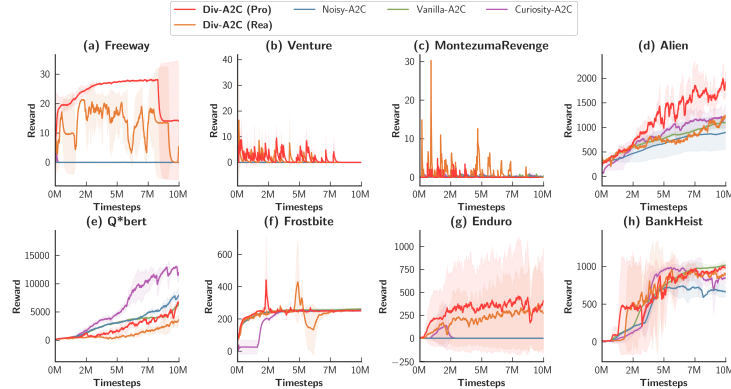

Figure 4: Comparison of learning curves for different A2C variants in Atari 2600.

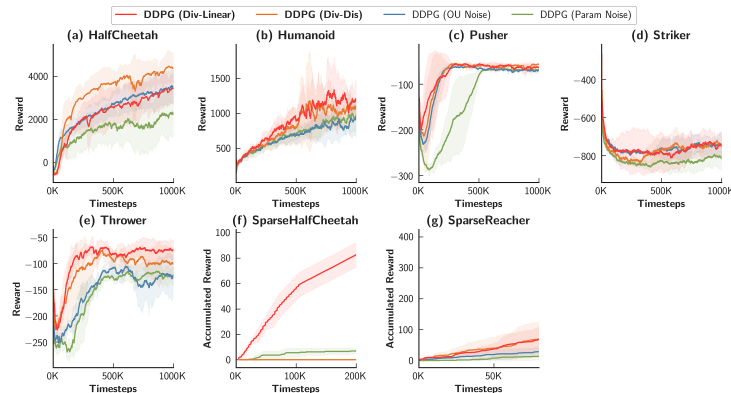

Figure 5: Comparison of learning curves for different DDPG variants in MuJoCo.

baselines. In summary, our methods bring an improvement in terms of scores and training efficiency for hard exploration games except *Montezuma's Revenge*, which is notorious for its complexity.

### 4.3.2 Easy Exploration Games

We show that the proposed diversity-driven exploration strategy can improve the training efficiency for easy exploration games as well. From the learning curves of *Enduro* and *BankHeist* presented in Figs. 3 and 4, it can be seen that our Div-DQN and Div-A2C agents learn significantly faster than the baselines in both games. They also show superior performance to the baselines for most of the time.

## 4.4 Performance Comparison in MuJoCo Environments

To evaluate the proposed methods in continuous control domains, we conduct experiments in MuJoCo environments with (1) deceptive rewards, (2) large state space, and (3) sparse rewards, and similarly plot the in-training median scores in Fig. 5. In Fig. 5, DDPG (Div-Linear) and DDPG (Div-Dis) represent Div-DDPG with linear decay and distance-based scaling, respectively. DDPG (OU Noise) and DDPG (Param Noise) stand for vanilla DDPG and parameter noise DDPG, respectively. We investigate and discuss how diversity-driven loss influences the agents' behavior in these environments, and demonstrate that our methodology does lead to superior exploration efficiency and performance.

### 4.4.1 Environments with Deceptive Rewards

Figs. 5 (a) and (b) plot the learning curves of Div-DDPG and vanilla DDPG in environments with deceptive rewards. In both cases, it is observed that our agents learn considerably faster, and end up with higher average rewards than the baselines. While vanilla DDPG often converges to suboptimal policies, Div-DDPG is able to escape from local optimum and find better strategies for larger payoffs. For example, in *Humanoid*, the baseline agent learns to lean forward for rewards, but at an angle that makes it easily fall down. Although our agent initially behaves in a similar way, it later discovers an alternate policy, and successfully walks forward for a much longer period without falling over.

Similarly, in *HalfCheetah*, Div-DDPG agent acts in the same way as the vanilla agent initially, but it ultimately learns to balance itself and moves forward swiftly. These results indicate that our method does help agents explore more states, increasing their chances to escape from suboptimal policies.

### 4.4.2 Environments with Large State Spaces

Figs. 5 (c), (d), and (e) plot the learning curves of Div-DDPG and DDPG in environments with large state spaces. It can be seen that our methods learn significantly faster than baseline methods in *Pusher*, *Thrower*, and *Strike*. In these environments, agents have to manipulate a robotic arm to push, throw, or hit a ball to goal areas, respectively. Even though these environments provide well-defined reward functions, it is still challenging for agents to learn feasible policies, as they have to explore the enormous state spaces. We observe that the baseline methods move the arms aimlessly, and rarely reach the goals. Moreover, the random perturbation in their behaviors makes it harder for them to push/throw/hit the ball to the goal. In contrast, without the interference of action noise during training, our methods can quickly learn to manipulate the arm correctly, and hence result in higher rewards.

### 4.4.3 Environments with Sparse Rewards

We redefine the reward functions of *Reacher* and *HalfCheetah* and create the *SparseReacher* and *SparseHalfCheetah* environments to investigate the impact of sparse rewards on the performance of Div-DDPG, parameter noise DDPG, and vanilla DDPG. In *SparseReacher*, a reward of +1 is only granted when the distance between the actuator and the target is below a small threshold. In *SparseHalfCheetah*, an agent is only rewarded with +1 when they move forward over a distance threshold. In Fig. 5 (f), we report the performance of each method in *SparseReacher*. While all of the methods are able to succeed in these environments, it can be noticed that Div-DDPG learns faster, and achieves higher average rewards. In Fig. 5 (g), we show the performance of each methods in *SparseHalfCheetah* with the distance threshold set to 15.0. It can be observed that Div-DDPG is the only method that is able to acquire a stable policy for this setting. In contrast, vanilla DDPG and parameter noise DDPG rarely exceed the distance threshold, and receive no reward most of the time.

From these results, we conclude that our methods equip agents with the ability to explore efficiently in continuous control environment, and achieve promising results in various challenging settings.

## 5  Related Work

As our diversity-driven exploration strategy relates to several prior works in RL, this section provides a comprehensive comparison with those researches in terms of objectives and implementations.

**Entropy regularization for RL.**   This line of works attempts to alleviate the premature convergence problem in policy search by regularizing the learning process with information-theoretic entropy constraints. In [25], the authors address this problem by constraining the relative entropy between old and new state-action distributions. Similarly, a few recent works [26, 27] propose to alleviate this problem by bounding the KL-divergence between prior and current policies. In terms of objectives, our method aims to improve the insufficient exploration problem in deceptive and sparse reward settings, rather than addressing the premature convergence problem in policy learning. Regarding implementations, both the above works and ours impose entropy-related constraints during learning. However, our method encourages exploration by maximizing the distance measure between the current and prior policies, instead of restraining their state-action distribution or KL-divergence.

**Maximum entropy principle for RL.**   This series of works aim to improve the performance of exploration under uncertain dynamics by optimizing a maximum entropy objective. In [28, 29], the authors construct this objective function by augmenting the reward function with policy entropy of visited states. Maximizing the expected entropy and rewards jointly encourages an RL agent to act optimally while retaining the stochasticity of its policy. This stochasticity in policy enhances the performance of exploration under uncertain dynamics. In respect of objectives, the works in [28, 29] and our approach all intend to ameliorate the performance and efficiency of exploration. However, our work focuses more on environments with deceptive and sparse rewards, rather than those with uncertain dynamics. In terms of implementations, our method maximizes the distance measure between old and new policies, instead of maximizing the expected entropy of an agent's policy.

To conclude, our method differs from the previous works in several fundamental aspects. It enhances the efficiency of exploration with a novel loss term. To our best knowledge, our work is the first one to encourage exploration by maximizing the distance measure between current and prior policies.

## 6 Conclusion

In this paper, we presented a diversity-driven exploration strategy, which can be effectively combined with current RL algorithms. We proposed to promote exploration by encouraging an agent to engage in different behaviors from its previous ones, and showed that this can be easily achieved through the use of an additional distance measure term to the loss function. We performed experiments in various benchmark environments and demonstrated that our method leads to superior performance in most of the settings. Moreover, we verified that the proposed approach can deal with sparsity and deceptiveness in the reward function, and explore in large state spaces efficiently. Finally, we analyzed the adaptive scaling methods, and validated that the methods do improve the performance.

## Acknowledgment

The authors would like to thank Ministry of Science and Technology (MOST) in Taiwan and MediaTek Inc. for their funding support, and NVIDIA Corporation and NVAITC for their support of GPUs.

## Footnotes

[1]https://github.com/openai/baselines

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
