[Supplementary Material]

# Supplementary Material for Diversity-Driven Exploration Strategy for Deep Reinforcement Learning

## S1   Choice of Distance Measure

The distance measure between the current policy and the old set of policies is determined by the specific network's form of outputs.

**DQN.**   In DQN, as the policy is implicitly defined by the Q-function, changes in the output Q-values do not necessarily reflect a change in the policy. This forbids us from calculating the distance directly by Q-values. To circumvent this issue, we apply the softmax function over predicted Q-values to get an action distribution (representing a policy). We then use the Kullback-Leibler (KL) divergence to measure the distance between action distributions.

**DDPG.**   As the outputs of DDPG are continuous, we simply adopt the mean squared error as the distance measure.

**A2C.**   Since A2C outputs a distribution over actions, we can directly use the KL divergence to measure the distance.

## S2   Training Detail

We follow the settings described in the respective original papers for vanilla DQN [1], A2C [2], DDPG [3], Parameter Space Noise [4], and NoisyNet [5]. Our Div-DQN, and Div-A2C are based on the network architecture of the vanilla models, and Div-DDPG follows the settings in [4].

**DQN.**   For Atari 2600, we search over as set of hyperparameters to optimize the performance of all methods on our codebases, as suggested in [6]. For each method, we apply the setting which yields highest overall performance. Note that we do not search $\epsilon$-greedy schedule for vanilla-DQN. The value of $\epsilon$ is always linearly annealed from 1.0 to 0.01. For gridworld tasks, we use a single layer MLP consisting of 64 hidden units followed by a ReLU activation.

**DDPG.**   For Sparse MuJoCo tasks, we follow the suggestions described in [4] to adjust the noise scale of baselines. We set the noise scale as 0.6 for Parameter Noise DDPG and vanilla-DDPG with action-uncorrelated noise. To prevent Div-DDPG from getting stuck in some deadlock pose, we also add action-uncorrelated noise with scale 0.6 for Div-DDPG.

**A2C.**   For Div-A2C, Noisy-A2C and vanilla-A2C, we collect the training rollouts by 16 concurrent workers.

## S3   Clipping Distance Measure

As described in Section 3.4, to prevent the instability issues during training, we clip the distance measure between $-c$ and $c$, where $c$ is a pre-defined constant value. We have different settings of c for Div-DQN, Div-A2C, and Div-DDPG, which are summarized in Table S1.

Table S1: Settings of $c$ for different models.

Value of $c$ for the Clipped Distance Measure

| | |
|---|---|
| Div-DQN | 2.5 |
| Div-DDPG | 0.2 |
| Div-A2C | 0.5 |

Table S2: Complete evaluation results of DQN/A2C variants in 21 Atari games.

| | Div-DQN | Noisy-DQN | DQN | Div-A2C-Pro | Div-A2C-Rea | Noisy-A2C | A2C |
|---|---|---|---|---|---|---|---|
| Alien | **2542.5** | 2270.2 | 1727.2 | **2475.8** | 1316.2 | 1347.7 | 1441.8 |
| Amidar | **738.35** | 530.73 | 359.52 | 336.78 | 273.09 | **388.71** | 351.98 |
| BankHeist | **1437.9** | 836.3 | 820.1 | 1087.0 | **1179.6** | 979.8 | 1165.0 |
| BeamRider | 5314.9 | **7565.66** | 4984.24 | 3301.74 | 2502.22 | 3086.82 | **3848.06** |
| Breakout | **397.24** | 156.13 | 238.64 | 380.85 | 319.97 | 386.71 | **388.22** |
| Enduro | **1053.0** | 968.68 | 428.7 | **810.39** | 623.75 | 0.0 | 0.0 |
| Freeway | **33.61** | 32.8 | 20.85 | **28.7** | 27.08 | 0.03 | 0.03 |
| Frostbite | **2695.7** | 385.0 | 1370.6 | **881.4** | 336.2 | 265.3 | 268.5 |
| Gravitar | 353.5 | 329.0 | **562.5** | 311.5 | **329.0** | 321.0 | 322.5 |
| MontezumaRevenge | 0.0 | **2.0** | 1.0 | 3.0 | **18.0** | 2.0 | 4.0 |
| Pitfall | -3.94 | **-0.28** | -5.37 | -32.92 | -32.28 | -34.01 | **-32.27** |
| Pong | **21.0** | 20.99 | 18.18 | 20.05 | **20.53** | 19.85 | 20.34 |
| PrivateEye | **2375.21** | 1109.01 | 1622.78 | **368.16** | 213.58 | 186.08 | 168.51 |
| Qbert | **14509.0** | 9737.5 | 8081.25 | 8604.0 | 4395.25 | **11320.75** | 9345.25 |
| Seaquest | 6113.3 | **6410.4** | 4205.4 | **1765.8** | 1748.0 | 1736.2 | 1717.2 |
| SpaceInvaders | 986.1 | **1190.55** | 1387.35 | 773.5 | 740.5 | 778.75 | **781.5** |
| Venture | **1025.0** | 15.0 | 15.0 | **12.0** | 9.0 | 0.0 | 0.0 |
| WizardOfWor | 1004.0 | **1620.0** | 1953.0 | 975.0 | 901.0 | 931.0 | **986.0** |
| Zaxxon | 6292.0 | **7594.0** | 4609.0 | **458.0** | 436.0 | 82.0 | 84.0 |

## S4 Adaptive Scaling

We have different adaptive and linear decay scaling settings for Atari 2600 and MuJoCo. In Atari2600, we set $\delta = 1.25$ for Div-DQN, $\delta = 0.25$ for Div-A2C since they have different magnitude on diversity loss. In MuJoCo, we set $\delta = 0.2$ for Div-DDPG.

## S5 Analysis of Adaptive Scaling Method

Adaptive scaling methods serve as a critical component of the proposed diversity-driven exploration strategy. Through the use of the adaptive scaling methods, the training progress can be stabilized, often leading to better overall performance and significant improvements over the naive linear decay method. Therefore, we investigate how different adaptive scaling methods influence the performance of our agents. For off-policy algorithms, it can be seen in Fig. 3, 4, and 5 that adaptive scaling method can draws better performance than linear-decay in *Venture* and *HalfCheetah*. This is due to the fact that exploration stops as $\alpha$ decreases to an extremely small value, making it hard for agents

to explore in huge state spaces, or escape from local optimum later in the training phase. On the other hand, both Div-DQN and Div-DDPG that employ the distance-based scaling method outperform their counterparts using the linear decay method, as the value of $\alpha$ is adaptively scaled. We further investigate whether the distance-based method is sensitive to the choice of the threshold value $\delta$. From the learning curves illustrated in Fig. S2 and S3, we find that the value of $\delta$ does not lead to much variation in performance for Div-DQN in most cases, but Div-DDPG is sensitive to $\delta$. Overall, we show that the distance-based method improves the stability of the learning progress, and is superior to the linear decay method. For on-policy algorithms, we focus on the comparison between distance- and performance-based methods, as the linear decay method is proven worse than distance-based method already. Fig. S1 plots the learning curves of Div-A2C using different scaling methods. We observe that in hard-exploration tasks such as *Freeway*, Div-A2C with distance-based method performs worse than the other, which agrees with our claim in Sec. 3.3. We further analyze the proactive and reactive performance-based methods. In Fig. S1, it can be observed that the proactive one demonstrates superior performance to the reactive one in most tasks. In particular, we find that reactive Div-A2C encounters the rewards in an early stage, but never learns an effective policy to exploit that knowledge to obtain higher average rewards. These results suggest that performance-based scaling method is indeed better than the distance-based one, and that the proactive strategy surpasses the reactive one.

Figure S1: Learning curves of Div-A2C with different scaling methods.

Figure S2: Learning curves of Div-DQN with different $\delta$.

Figure S3: Learning curves of Div-DDPG with different $\delta$.