[Reviews · NeurIPS 2018]

Reviewer 1



In this paper, the authors propose to improve exploration in deep RL algorithms by adding a distance term into the loss function. They show that adding this term provides better results that not doing so. After rebuttal: The authors did a much better job explaining their work in the rebuttal, so I'm now convinced that they have a contribution. I'm now more inclined in favor of this paper, but the authors will have to explain much more carefully what they are doing (included a better presentation of the formalism) and how it is positionned with respect to the literature. I keep the rest of the review as it was. The first issue with this paper is that the authors seem to completely ignore the very large "relative entropy" or "entropy-regularized" body of work, as they cite none of the relevant work in the domain (e.g. Haarnoja et al., Neu et al., Peters et al. etc.). To me, this weakness is serious enough to reject the paper, as the authors fail to properly position their contribution with respect to their natural field. From the relative entropy or entropy-regularized perspective, the contribution of the authors, whose technical part is located in Eqs. (5), (6) and (7) could be described as performing entropy-regularized RL, but using an average over the n=5 previous policies rather than just the previous one. This characterization is much simpler than the one they propose, and if it happens to be too simplified, the authors could explain what are the differences and why they matter. By the way, from the above perspective, the formal writing of the expectation in (5), (6) and (7) is partly wrong or at least lacking rigor. Again, since this is the main contribution, a much clearer and rigorous description is mandatory. Also, the fact that using relative entropy improves exploration is now well-known, thus the empirical results are not surprising to me at all. An empirical comparison between the authors framework and other entropy-regularized methods could have delivered much more interesting messages. Other weaknesses contribute secundarily to my negative evaluation about this work: - Section 3.3 and 3.4 are full of un-principled tricks and hyper-parameters - the influence of those tricks and hyper-parameters is not properly studied - results in Fig.5 contradict the literature: the authors found that DDPG with OU noise outperforms DDPP with parameter noise, whereas several papers have found the contrary. I'm afraid the empirical study lacks a lot of evaluation rigor: no number of seeds is specified, statistical significance is not assessed, etc. In the background section, the paragraphs are too short and not so well written, they do not help much. Finally, the paper suffers from a few more local formatting or writing problems, but this does not contribute significantly to my negative evaluation: - Subfigs (a) to (c) could be above subfigs (d) to (f). - "sufficient enough" is redundant ;) - l. 48 to 52: two sentences that are quite redundant - l.92: an experience replay => a replay buffer? typos: l. 7: preventing => prevents l. 33: its simpliciy => their l. 113: compared with => to l. 129 eq. (4) => Eq. l. 263: of all of the models => of all models References: V. et al. Mnih should be V. Mnih et al. Many references suffer from the same problem. Supplementary: the content of S1 should stay in the main text rather than being in an appendix

Reviewer 2



[Summary]: The paper proposes a formulation for incentivizing efficient exploration in deep reinforcement learning by encouraging the diversity of policies being learned. The exploration bonus is defined as the distance (e.g., KL-divergence) between the current policy and a set of previous policies. This bonus is then added to the standard reinforcement learning loss, whether off-policy or on-policy. The paper further uses an adaptive scheme for scaling the exploration bonus with respect to external reward. The first scheme, as inspired from [Plappert et.al. 2018], depends on the magnitude of exploration bonus. The second scheme, on the other hand, depends on the performance of policy with respect to getting the external reward. [Paper Strengths]: The paper is clearly written with a good amount of details and is easy to follow. The proposed approach is intuitive and relates closely to the existing literature. [Paper Weaknesses and Clarifications]: - My major concerns are with the experimental setup: (a) The paper bears a similarity in various implementation details to Pappert et.al. [5] (e.g. adaptive scaling etc.), but it chose to compare with the noisy network paper [8]. I understand [5] and [8] are very similar, but the comparison to [5] is preferred, especially because of details like adaptive scaling etc. (b) The labels in Figure-5 mention that DDPG w/ parameter noise: is this method from Plappert et.al. [5] or Fortunato et.al. [8]. It is unclear. (c) No citations are present for the names of baseline methods in the Section-4.3 and 4.4. It makes it very hard to understand which method is being compared to, and the reader has to really dig it out. (d) Again in Figure-5, what is "DDPG(OU noise)"? I am guessing its vanilla DDPG. Hence, I am surprised as to why is "DDPG (w/ parameter space noise)" is performing so much worse than vanilla DDPG? This makes me feel that there might be a potential issue with the baseline implementation. It would be great if the authors could share their perspective on this. (e) I myself compared the plots from Figure-1,2,3, in Pappert et.al. [5] to the plots in Figure-5 in this paper. It seems that DDPG (w/ parameter space noise) is performing quite worse than their TRPO+noise implementation. Their TRPO+noise beats the vanilla TRPO, but DDPG+noise seems to be worse than DDPG itself. Please clarify the setup. (f) Most of the experiments in Figure-4 seems to be not working at all with A2C. It would be great if authors could share their insight. - On the conceptual note: In the paper, the proposed approach of encouraging diversity of policy has been linked to "novelty search" literature from genetic programming. However, I think that taking bonus as KL-divergence of current policy and past policy is much closer to perturbing policy with a parameter space noise. Both the methods encourage the change in policy function itself, rather than changing the output of policy. I think this point is crucial to the understanding of the proposed bonus formulation and should be properly discussed. - Typo in Line-95 [Final Recommendation]: I request the authors to address the clarifications and comments raised above. My current rating is marginally below the acceptance threshold, but my final rating will depend heavily on the rebuttal and the clarifications for the above questions. [Post Rebuttal] Reviewers have provided a good rebuttal. However, the paper still needs a lot of work in the final version. I have updated my rating.

Reviewer 3



This paper is about exploration strategies for deep reinforcement learning. They present approach based on encouraging the agent's policy to be different from its recent policies. This approach is simple and easy to implement, and they show variations of it implemented within DQN, A2C, and DDPG. The show nice results, particularly in the sparse reward versions of the mujoco tasks. This work is related to previous works on exploration in deep RL such as the work on noisy nets or Bellemare's work on using density estimators to drive exploration. However this approach is simpler to implement and has better empirical results. The strengths of the paper are its originality and significance. The ideas in the paper are original, and since they show good results and are easy to implement across a set of deep RL algorithms, I believe they have the potential to be signfiicant and taken up by many deep RL practitioners. The paper is well-motivated and the experiments are detailed (multiple variations of the idea applied to 3 different algorithms across many different domains). There are a few things in the implementation and experiments that are unclear, however. The paper would be much improved with these issues clarified. Line 139 says that the agent is ensured to act consistently in the states its familiar with, as pi and pi' will yield the same outcomes for those states. I'm not sure what would ensure this to be the case. For A2C, the authors state that they maintain the n most recent policies. What does this mean exactly? They keep copies of the last n networks? The n most recent means the n networks after the last n updates? For the distance-based scaling, it seems that as the policy converges towards optimal, the D distance will go towards 0 and the alphas will continually increase. This doesn't result in instability in the algorithm? For the performance based policies, you measure the performance P(pi') over five episodes. How does this evaluation happen? Do you run this separately or was each pi' already kept unchanged for 5 episodes? There's a few minor errors in the paper: - Line 238 is the first time Div-DQN is introduced with out explanation of what it stands for. - None of the text explains or compares to the Curiosity-DQN results in the plots. - It's not obvious what the Div-DQN algorithms in the plot legends refer to (e.g. what is Div-DQN(Linear)?) - The refernece list format is broken, many of the papers show up with the authors as {first initial} et al. {last name}